# Statistical Conceptualisation of Mood Instability: A Systematic Review

**DOI:** 10.3390/brainsci15101059

**Published:** 2025-09-29

**Authors:** Iona Cairns, Kim Wright, Gordon Taylor, Bryher Mehen, Ruth Anning

**Affiliations:** 1Department of Psychology, Faculty of Health and Life Sciences, University of Exeter, Washington Singer Labs, Perry Road, Exeter EX4 4QG, UK; ic408@exeter.ac.uk (I.C.); b.k.mehen@exeter.ac.uk (B.M.); r.anning@exeter.ac.uk (R.A.); 2University of Exeter Medical School, University of Exeter, St Luke’s Campus, Heavitree Road, Exeter EX1 2HZ, UK; g.j.taylor@exeter.ac.uk

**Keywords:** mood instability, affective variability, experience sampling

## Abstract

**Background/Objectives**: Our understanding of mood instability as a clinically important feature of many psychiatric conditions has been increasing over the last decade, but there remains a lack of clarity around the optimal ways to calculate mood instability in real time. We conducted a systematic review in order to describe the statistical methods used in studies investigating mood instability that collected mood data using ESM (Experience Sampling Methodology). **Results**: From a total of 229 papers, we found 15 discrete statistical methods were used a total of 319 times. In 76 (33%) studies, more than one statistical method was used, and 39 (17%) studies employed distinct statistical methods for particular aspects of affect dynamics. **Conclusions**: Based on our findings, we recommend standardisation of statistical methods to strengthen future research on mood instability and ultimately support better clinical outcomes for individuals with mood difficulties.

## 1. Background/Objectives

Mood instability has been increasingly recognised as a core component of numerous psychiatric disorders, including but not limited to borderline personality disorder, bipolar disorder, and various anxiety disorders [1]. Despite its prevalence and impact, mood instability has historically been underemphasized in psychiatric assessment and treatment paradigms and often deprioritised in clinical frameworks [2]. This has undoubtedly limited a deeper understanding of mood instability, its implications for patient outcomes, and the development of targeted therapeutic strategies.

Clinically, mood instability holds significant importance and is associated with a range of poor outcomes across diagnostic categories, including impaired functioning and higher distress [3], increased healthcare service utilisation [4], a negative impact on interpersonal relationships [5], and is among the strongest predictors of suicidal behaviour [6]. Despite its evident clinical relevance, a lack of clarity regarding its definition, measurement, and analysis has hindered the ability to develop standardised approaches to its study and treatment [7].

One key challenge lies in the inconsistent conceptualisation of mood instability itself across the literature. Definitions often overlap between related constructs such as affective lability, emotional dysregulation, and mood variability [8]. For instance, affective instability has been described as extreme shifts in mood, heightened reactivity to environmental stimuli, and transient emotional fluctuations [9], whereas other researchers emphasise frequent or extreme mood changes over time [1]. These differing terminologies have created ambiguity, complicating efforts to establish diagnostic criteria, investigate underlying mechanisms, and design effective interventions. A list of mood constructs, selected concise definitions, and commonly applied analysis methods (Table 1) should serve as a first step towards reducing this ambiguity. Though the literature has many overlapping definitions and conceptualisations of affect dynamics, these definitions and measures were selected on the basis of being up to date and relevant to our review aims.

In this review, we have used the term “mood instability” (MI) broadly to include studies examining fluctuations in mood across time. We acknowledge, however, that alongside the inconsistencies outlined above, the literature often distinguishes between mood variability—referring to typical, non-pathological fluctuations in mood—and mood instability, which is often used to describe dysregulated or pathological mood changes.

The terms used in the literature to refer to mood and affect instability include, but are not limited to, instability (the quality of being unstable), variation (the difference or deviation in mood), inertia (mood states being resistant to change), valence (the extent to which an emotion is negative or positive), and intensity (degree of emotion). These terms also imply important differences statistically. For example, variability and instability are terms which have been used interchangeably in the literature for many years [18,19], and we can elucidate the importance of their differentiation using this metaphor based on Ebner-Priemer and colleagues (Figure 1) [20]. Imagine two students go on separate two-week holidays. Student A has one solid week of sunshine and a second week of consistent cloud and rain. Student B, meanwhile, experiences alternating sunshine and rain for the entire two weeks. The variability of the weather for students A and B was the same, but the stability of the weather was very different. When it comes to mood, two individuals may have very different personal experiences of mood changes, which the construct of ‘variability’ alone may be insufficient to capture.

These conceptual differences in how we think about mood instability imply a need for different means of characterising them statistically, and this is reflected in some published studies exploring mood instability. For example, while variability encompasses general dispersion of scores, instability is said to include temporal aspects such as frequency, amplitude, and sequence of mood changes. Jahng and colleagues [1] proposed metrics such as mean squared successive differences (MSSD) and probability of acute change (PAC) to better capture this temporal instability, whereas standard deviation (SD) is often used to capture variability.

Valid means of calculating mood instability are particularly important given recent developments in affect measurement. Traditional approaches to mood assessment often relied on retrospective self-report questionnaires, which are prone to recall bias and fail to capture dynamic mood changes [2]. Ecological momentary assessment (EMA) and other intensive longitudinal data collection methods have emerged as promising alternatives, offering real-time, ecologically valid insights into mood dynamics [21]. These are methods where repeated data points are collected in real-time. The major advantage of using this type of methodology is the ability to investigate mood with high ecological validity in the real world outside of an interview room. For the purposes of this review, we focused upon studies that used EMA or similar data collection methods.

With no current consensus on the most appropriate statistical methods, this review aims to systematically examine how mood instability has been calculated from ESM data within the extant literature. We aim to contribute towards the development of a taxonomy of statistical methods in mood instability research, ultimately improving the reliability and interpretability of findings.

## 2. Methods

### 2.1. Eligibility Criteria

This systematic review focused on quantitative studies employing a prospective design to analyse mood instability using data collected over multiple time points. This review was pre-registered on PROSPERO (registration number: CRD42023477875). Studies were included if they employed EMA or similar methodologies to capture real-time mood data, measured mood instability, variability, or related constructs over time (including mood and affect measured in clinical conditions), used prospective designs with at least three time points of data collection, had intervals no longer than one week, and reported on human populations with no restriction on age or clinical status. Studies were excluded if they relied exclusively on retrospective self-report measures, were not published in English and lacked available translations, or focused solely on cross-sectional data or single time point analyses. Only studies published after 1950 were included.

### 2.2. Information Sources

Three databases were searched: EMBASE, MEDLINE, and PsycINFO. Keywords were developed to capture both conceptual and methodological aspects of mood instability, including terms such as “emotional instability,” “affective lability,” “mood variability,” and “ecological momentary assessment.” The initial searches took place in November 2023, with further searches in January 2025. We acknowledge the limitations in our searches due to the overlapping terminology used to describe both mood instability and EMA data collection. Studies may have used a repeated-measures data collection methodology without naming the process as EMA or mood monitoring. Hand searching was conducted in March 2025 to identify relevant studies which may not have used common terminology to refer to EMA but nonetheless fit our criteria. We did this by examining references from relevant review papers that explored methodologies used in mood research (see Appendix A for more details).

### 2.3. Search Strategy

The search was structured as follows:Conceptual Terms: “emotion$ instability,” “affect$ instability,” “mood instability,” etc.Methodological Terms: “ecological momentary assessment,” “experience sampling,” “mood monitoring,” etc.Combined Search: Boolean operators were used to link conceptual and methodological terms (e.g., “mood instability AND ecological momentary assessment”). Full Search Terminology (see Appendix A).

### 2.4. Full Searches

Exp emotional instability.‘emotion$ instability’ or ‘emotion$ unstable’ or ‘emotion$ labil$’ or ‘emotion$ dysregulat$’ or ‘emotion$ variab$’ or ‘emotion$ fluctuat$’.‘affect$ instability’ or ‘affect$ labil$’ or ‘affect$ dysregulat$’ or ‘affect$ variab$’ or ‘affect$ fluctuat$’.‘Mood instability’ or ‘mood labil$’ or ‘mood dysregulat$’ or ‘mood variab$’ or ‘mood fluctat$’.(1 or 2 or 3 or 4).Exp ecological momentary assessment.‘experience sampling’ or ‘experience sampling method$’.Mood monitoring.(6 or 7 or 8).(5 AND 9).

Limits.

English Language1950—current.

### 2.5. Selection Process

Study screening and selection were conducted using the web-based Covidence systematic review software (Covidence, Veritas Health Innovation, Melbourne, Australia; available at www.covidence.org). Two reviewers independently assessed 20% of titles and abstracts to ensure consistency, resolving disagreements through discussion or referral to a senior researcher. This process was repeated until inter-rater agreement exceeded 80%, after which remaining titles and abstracts were reviewed by a single reviewer. This process was repeated for the full texts. During the single reviewer process, uncertainties were also resolved through discussion with a senior researcher.

### 2.6. Data Collection Process

Covidence software was used for data extraction. This software allowed a personalised template to be created with the preferred data items. This was performed by one researcher and checked by a second researcher for a selection of 20% of the papers. Any uncertainties were resolved in discussion with a senior researcher. Data was exported from Covidence to a CSV file for analysis.

### 2.7. Data Items

Data extraction included the following items: publication details (e.g., title, country, year, institution), study design and methodology (e.g., sample size, EMA measures, data collection method and frequency, time period, number of mood items, number of data points, type of population), terminology and definitions of mood instability (discrete constructs of mood, e.g., variability, valence, instability), statistical methods and number of methods used, whether mood instability was a predictor or independent variable, software used, any comparison to another variable, and main outcomes and comments about mood instability. EndNote and Zotero were employed for citation management.

### 2.8. Study Risk of Bias Assessment/Effect Measures

In this review, the outcome of interest was the design choice made by the research team in each study, namely the method used to calculate mood instability. Because we were not examining standard outcomes we did not carry out a risk of bias assessment. In-depth assessment of the quality of review evidence was also deemed non-applicable for the same reason. We carried out a simplified quality assessment by categorising studies according to the frequency of EMA: high (≥3 per day), moderate (1–2 per day), low (<1 per day), or unclear/not reported.

### 2.9. Synthesis Methods

Extracted data were organised into groups to help identify patterns and differences in the data. For example, the type of sample was grouped into three umbrella categories: clinical, non-clinical, and clinical versus healthy controls. The umbrella categories were further subcategorised based on the type of sample, for example, employees in a company or patients with anorexia. We categorised other variables in the same way, grouping and refining the raw data into categories. This allowed us to look at patterns in the choice of calculation methods alongside the rest of our data. The same process was followed for the statistical methods themselves, looking in depth at the exact method used and grouping them to allow uniform terminology across our review. We also categorised the exact methods and evaluated them based on their ability to capture relevant dimensions of mood dynamics, such as variability, instability, and temporal dependencies. We performed chi-square tests of independence to examine choice of methods in relation to other aspects of the studies.

## 3. Results

We identified a total of 926 studies from our database searches and hand searches from reference lists of relevant systematic reviews (see PRISMA flow diagram Figure 2). We removed 462 duplicates manually in EndNote and Covidence. After title and abstract screening, 95 further studies were removed. In total we screened 367 studies for eligibility; 28 were removed as we were unable to retrieve full texts, 88 studies were excluded because they had the wrong outcomes (did not report a mood instability calculation method), and a further 22 studies were removed as they did not meet our study design criteria. The remaining 229 studies were included in our review.

### 3.1. Study Characteristics

Studies were conducted in 24 countries: Australia, Austria, Belgium, Canada, China, Denmark, Estonia, Finland, France, Germany, Iceland, Ireland, Israel, Italy, Japan, New Zealand, the Republic of Korea, Singapore, Switzerland, Taiwan, the Netherlands, the UK, and the USA. The total number of participants for each study ranged from 1 to 2125. The time period adopted in each study to monitor mood ranged from 1 to 1540 days. The frequency of mood monitoring ranged from every 15 min to once a week (48 to 0.14 times per day). After removing overlapping terminology, a total of 15 different statistical methods were identified as being utilised to calculate mood instability across the 229 papers included. These methods were used a total of 319 times. The number of methods used ranged from 1 to 5 (mean 1.5, SD 0.96). Overall, 27 different terminologies were found to describe the 15 methods (see Appendix A).

We looked at the percentage use of each method across all the EMA studies and found that three methods accounted for 76% of the methods used in total (Figure 3). These methods were standard deviation (25%), mean squared successive difference (36%), and multi-level model (15%). (We include the percentage uses of these methods in Appendix A, including all the different terminologies Appendix A).

### 3.2. Statistical Methods Chosen over Time

The earliest paper included was published in 1973, and the most recent in 2024. We found the use of methods changed over time, with the introduction and increasing use of particular methods such as mean squared successive difference (Figure 4).

### 3.3. Type of Sample

We classified the types of populations studied by the 229 papers and found 73 (32%) investigated mood instability in a clinical sample, e.g., individuals with bipolar disorder. We found 111 (48%) recruited participants from a non-clinical population, such as employees in a company or the general population, and 42 (18%) studied a clinical sample compared with a comparison sample without the condition. We also found 160 (70%) of the studies investigated mood instability compared with another variable, for example, mood instability versus depression symptoms, and 64 (28%) studies compared mood instability between two or more groups, for example, patients with bipolar disorder and borderline personality disorder.

We investigated whether the statistical methods chosen differed depending on the type of sample being investigated. A chi-square test was performed to examine the relation between type of sample and use of MSSD, SD, and MLM (Table 2). These methods were chosen because they were the most frequently used. The relation between these variables was significant with a weak association, X^2^ (1, N = 235) = 14.5, *p* < 0.01, *V* = 0.18. Deconstructing the significant effect using further chi-square tests (see Appendix A), this difference appears to reflect greater use of MSSD rather than SD and MLM in studies involving clinical samples compared to studies of non-clinical samples only.

We explored whether multiple statistical methods were used to conceptualise mood instability in the literature. We found 153 (67%) studies used only one statistical method in their analysis and 77 (34%) studies used more than one. We used a chi-squared test of independence (Table 3) to find out if the number of methods used differed according to the type of sample and found there to be no significant difference X^2^ (1, N = 227) = 1.18, *p* = 0.28, *V* = 0.07).

A large number of studies (*n* = 190, 83%) used mood instability as their dependent or outcome variable. A much lesser proportion (*n* = 34, 15%) used MI as an independent/predictor variable. Only 5 (2%) studies used mood instability as both the predictor/independent variable and the outcome variable. Using a chi-square test, we found no statistical difference X^2^ (1, N = 199) = 2.89, *p* = 0.08) between the choice of statistical methods used to calculate MI when used as an independent or dependent variable.

### 3.4. Constructs of Mood

Across the 229 papers, 22 discrete affect dynamics were mentioned, which are presented as a word cloud (Figure 5), a visual depiction of the relative use of the different affect dynamics used across the studies. The larger the word, the more times this word was found throughout the studies. As can be seen, the most used term or affect dynamic reported on was ‘variability’, followed by ‘instability’.

We found 123 (54%) of the 229 studies mentioned only one aspect or construct of mood, while 107 (47%) of the remainder separated mood out into separate affective dynamics, e.g., reactivity, variability, instability, and inertia [22]. A total of 77 (34%) studies used more than one statistical method, yet only 39 of those (51% of this subset; 36% of the 107 reporting on separate affective dynamics) matched different statistical methods to specific aspects of mood or affect dynamics. For example, 36 studies mentioned the different aspects of variability and instability, yet only 20 (56% of this subset) treated them differently statistically. Figure 6 shows the use of different statistical methods for specific aspects of mood instability. For the full list of dynamics and statistics used, see Appendix A.

## 4. Discussion

In this review we were able to characterise how mood instability has been calculated in the current literature. This is in response to an acknowledged lack of consensus on the optimal ways of defining, measuring, and analysing mood instability [2,22,23,24]. In 2019, Faurholt-Jepsen and colleagues [23] published an important set of guidelines for the standardisation of reporting and analysing mood as well as describing the empirical evidence for a lack of clarity in the literature. Our review should serve as support for this goal of standardisation by specifically focusing on the statistical methods applied to mood data.

We appreciate that in standardising statistical methodology, we must overcome the challenge of highly diverse mood experiences between subjects and populations. Context will remain important when choosing the right methods for this reason, and the type of data, population, and situation will no doubt influence methods chosen. We also note that types of variability will not be consistent across disorders or individuals, but rather where there are identifiable patterns, we argue it is desirable that there is a standardised means of analysing the data according to the named pattern of instability. We plan to carry out further research about the application of these methods and their clinical relevance.

Although advanced approaches such as MLM provide powerful tools for modelling variability, their complexity and data requirements may limit their feasibility in clinical settings where real-time application is desired. In contrast, simpler indices such as SD and MSSD, while more limited in scope, have the advantage of being straightforward to compute and interpret, making them more practical candidates for clinical use. Ultimately, the choice of method should balance statistical rigour with feasibility in the intended applied context.

Though many calculation terms have been used with overlapping meanings, we found 15 clearly distinct approaches to calculating mood instability from ESM data. Variations in within-person standard deviation and mean squared successive differences were by far the most used methods and tended to be overrepresented in studies that investigated clinical populations. We observed that over the past few decades a historic preference for standard deviation has been supplemented by a growth in the use of new and more complex methods. While increasingly sophisticated methods allow for more nuanced exploration of variability, each commonly used approach carries important limitations that must be considered in EMA research. For example, mean squared successive difference is valuable for capturing temporal instability but assumes equal spacing between assessments, an assumption that is often violated in naturalistic EMA designs. Standard deviation (SD), although straightforward and widely interpretable, summarises dispersion without regard to temporal order, thereby obscuring patterns of fluctuation. Both methods assume normally distributed data, which may not always be the case, for example, if the individual tends to be in one of two mood states. Multilevel modelling provides flexibility in partitioning within- and between-person variance and testing predictors of variability, yet it requires careful specification of covariance structures and sufficiently large samples to produce stable estimates. Thus, the choice of method should be guided not only by the research question but also by the statistical appropriateness given the design and sampling features of the EMA data.

By looking at the use of methods by year, we were also able to identify certain patterns emerging in terms of the use of specific statistical methods to investigate specific affect dynamics. Guided by Sperry’s 2019 study [22], in which inertia, reactivity, instability, and variability were identified as primary affect dynamics, we examined use of this or similar categorisations within our study set. While almost half of the studies included in this review pointed at more specific aspects or affect dynamics within ‘mood instability’, such as those above, only around a third of these treated them differently in any statistical sense. Within these, however, there was broad agreement regarding which statistical method ‘matched’ which affective dynamic construct. In terms of developing a suggested framework for methodologies, an important future step is to apply these methods to existing data sets to examine their predictive and divergent validity, particularly with regard to associations with relevant individual difference variables and past and future clinical and functional outcomes. At this current stage, our recommendations are based more upon theoretical rather than empirical distinctions; as such, the findings of our review support use of Sperry’s 2019 identified affect dynamics [22].

Over the last decade increasing attention has been given to mood instability as an outcome measure within RCTs [2,23] because it is increasingly looked upon as an important potential target for both pharmacological and psychological therapies. Our review provides a description of the methods used in the literature to date, paving the way for future work to determine which are reliable and valid means of representing the particular constructs of interest.

Beyond research, standardised variability metrics have potential relevance for clinical and digital health applications. For example, methods such as SD and MSSD could be embedded into digital health apps to provide patients with accessible summaries of their daily emotional variability, which may support self-awareness and self-management. Clinicians might use these same indices to monitor changes in variability over time, identifying patterns that might signal relapse, treatment response, or heightened risk. These potential applications in clinical care remain speculative and would require further work to ensure beneficial use in practice.

We note that this review focused solely on *analysis* of mood instability, and just as relevant to standardisation in mood instability research will be work on *measurement* of mood. We observed substantial between-study variation in the number and content of the mood measurement items presented as well as the frequency and time period over which mood was monitored, the latter of which we examined as a simplified quality indicator. The majority of studies employed a high-frequency design (≥3 per day), consistent with the aim of capturing fine-tuned temporal dynamics. However, a small proportion relied on moderate or low-frequency sampling, and some did not report frequency at all. Standardisation of reporting EMA frequency and greater consistency in sampling intensity could improve the comparability and interpretability of future research.

Another limitation of this review is that eligible studies may have been missed due to their use of terminology not included in our searches, despite hand-searching of review reference lists. While we did not carry out a risk of bias assessment on each study as per PRISMA 2020, we acknowledge some general risks of bias across the studies included in this review. For example, we note excluding papers not published in English may have biassed our findings, especially as EMA data collection is increasingly used globally. There also may have been sampling bias and publication bias (studies with null findings may not have been published).

## 5. Conclusions

Our review highlights the wide range of statistical approaches used to characterise mood instability and emphasises both the progress made and the challenges in reaching a methodological consensus. While options such as SD, MSSD, and MLM each offer unique strengths, their limitations and feasibility can differ across research and clinical contexts. Standardisation of analytic approaches will be essential for improving comparability, supporting clinical translation, and advancing mood instability as a meaningful outcome in both research and practice. Future research should focus on evaluating the reliability, validity, and clinical feasibility of these methods to determine their suitability for application in mental healthcare.

## Figures and Tables

**Figure 1 brainsci-15-01059-f001:**
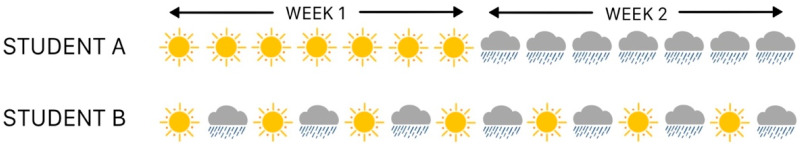
Diagram to show weather patterns on two students’ summer holidays, based on Ebner and colleagues [12]. The top row shows student A’s weather: sunshine the first week and clouds and rain the second week. The bottom row shows Student B’s week of weather: intermittently sunny and rainy.

**Figure 2 brainsci-15-01059-f002:**
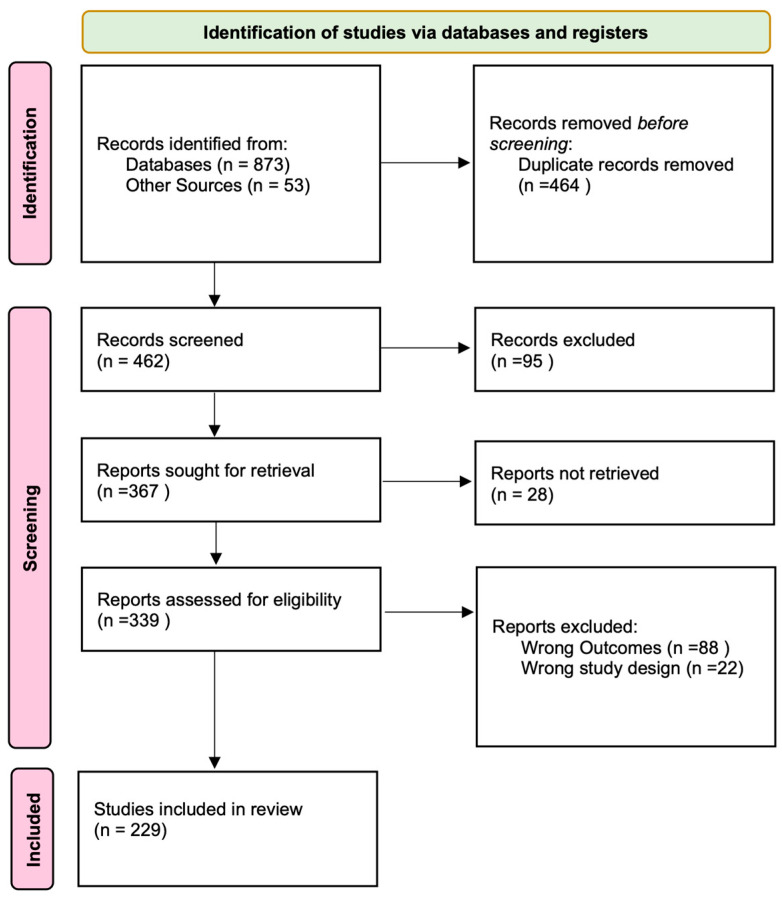
PRISMA flow diagram to show our process from record identification to study inclusion.

**Figure 3 brainsci-15-01059-f003:**
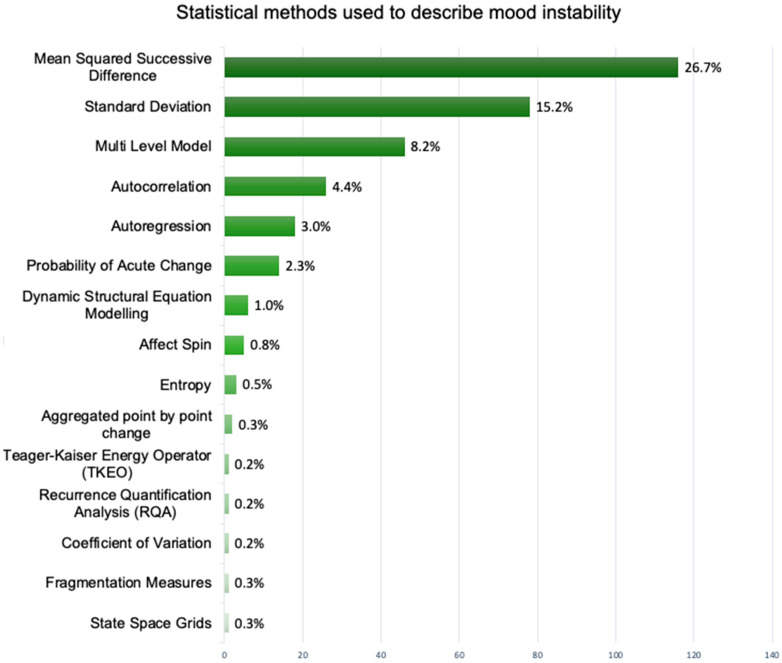
Chart to show use of statistical methods used to conceptualise mood instability across all 229 included studies.

**Figure 4 brainsci-15-01059-f004:**
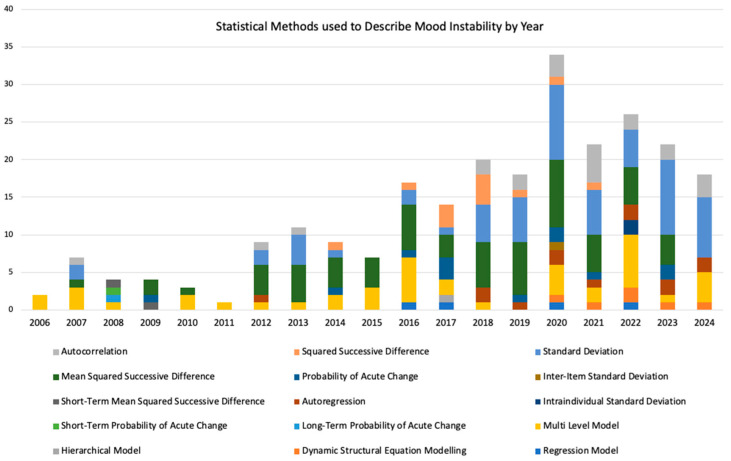
Statistical methods used to describe mood instability over time. Between 1973 and 2006, there were 2 or fewer studies per year, and they predominantly used standard deviation to describe mood instability.

**Figure 5 brainsci-15-01059-f005:**
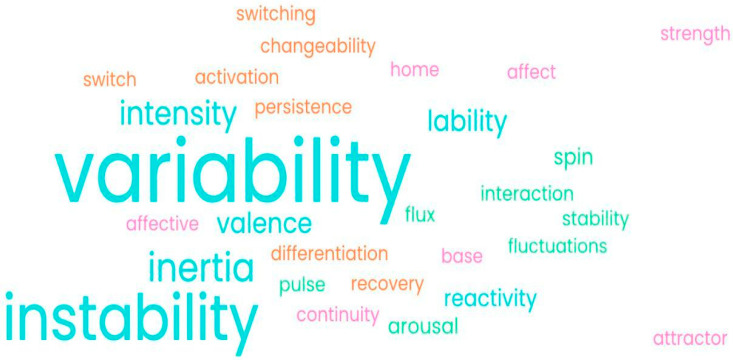
Word cloud showing the relative use of terminologies to describe mood instability. All mood constructs of interest found across the included studies are shown in various sizes depicting how frequently the constructs were used. Variability being the most commonly used, it appears the largest.

**Figure 6 brainsci-15-01059-f006:**
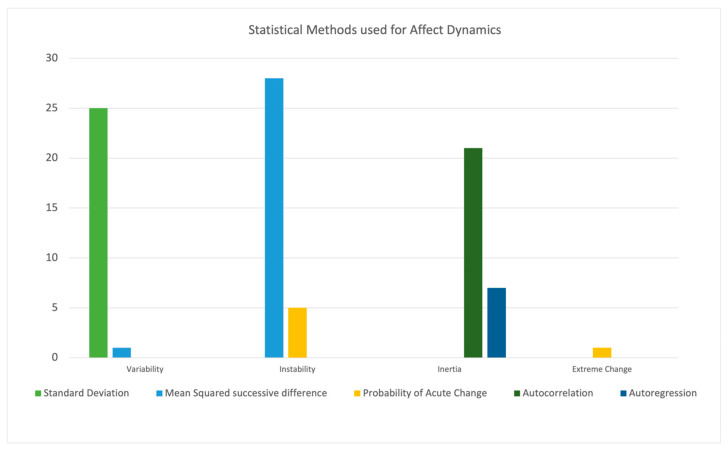
Bar chart to show statistical methods used to calculate specific affect dynamics (variability, instability, inertia, and extreme change). The chart shows more than one method was used to analyse the same construct of mood. For example, instability was analysed by both mean squared successive difference (MSSD) and probability of acute change (PAC) with different frequencies.

**Table 1 brainsci-15-01059-t001:** Table to show concise definitions and corresponding measures from prior literature, including references.

Mood Construct	Definition	Methods Commonly Used	References
Variability	Overall amplitude of affective changes (i.e., how much affect deviates around its mean level)	Within-Person Standard deviation	[10]
Instability	Variability of mood over time	Mean squared successive difference (MSSD) or root mean squared successive difference (RMSSD)	[10]
Inertia	Resistance to change; the tendency for affect to *persist* from one timepoint to the next	Autocorrelation	[11,12]
Intensity	Strength of mood/affective state regardless of valence.	Measured via distance from neutrality or baseline	[12]
Valence	The quality of an affective experience in terms of how pleasant vs. unpleasant it is	In dimensional models (circumplex), valence is one axis (pleasantness/activation as two axes).	[12,13]
Arousal	The degree of physiological or subjective activation/excitement vs. calmness (ranging from feeling quiet to active).	measured by self-report (low to high activation)	[12,13]
Flux/Fluctuations	Degree of within-person variation in specific affect dimensions (valence, arousal, or quadrants).	within-person standard deviations (SD) of affect ratings over time;	[12,14]
Pulse	Pulse refers to variability in *intensity of affective states*, regardless of their valence or activation direction. (intensity variability)	within-person standard deviation of the distance of each affect state from neutral (in valence-activation space). So first compute each time-point’s “distance” from neutral, then see how much that varies.	[15,16]
Spin	within-person fluctuations in affect pleasantness and activation potential.	Computed via the angular dispersion of core affect trajectories: take each moment’s core affect (valence & activation), compute its angle relative to some origin, then compute variability (SD) of those angles across time (around that person’s mean angle). Higher spin → experiences more qualitatively different emotions over time.	[15,16,17]
Reactivity	change in mood in response to events (often modelled as slope/interaction terms in multilevel models	Typically modelled as slope/interaction. Often multilevel regression of affect on stressor.	[10,11]

**Table 2 brainsci-15-01059-t002:** Chi-square distribution table showing clinical and non-clinical samples and use of MSSD, SD, and MLM.

	All MSSD	SD	MLM
Non Clinical	42	46	27
Clinical	47	17	10
Clinical and Healthy controls	26	13	7

**Table 3 brainsci-15-01059-t003:** Chi-square distribution table showing clinical and non-clinical samples and use of different statistical methods.

	Only One Method	More than One Method
clinical	48	26
Non clinical	78	33
Clinical vs. non clinical	26	16

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
