# Peer review of "Statistical Conceptualisation of Mood Instability: A Systematic Review"

_brainsci, 2025, doi:10.3390/brainsci15101059_

Round 1

Reviewer 1 Report

Comments and Suggestions for Authors

This is an interesting review of mood instability in clinical psychiatric care.
Please provide additional explanation for the following points:

  1. 1. Page 2, Figure 1
    The periodicity of Students A and B is depicted. Individual differences between subjects and clients are highly diverse, making standardizing statistical methods difficult. Please provide additional explanation for this difficulty.

  2. 2. Statistical methods used in academic papers tend to be those that reveal more significant differences. Furthermore, statistical methods vary depending on the characteristics of the subjects and the specifics of the clinical research. For these reasons, standardizing and unifying statistical methods is considered difficult. Please provide additional explanation for this point as well.

Reviewer 2 Report

Comments and Suggestions for Authors

This manuscript presents a systematic review examining statistical methods used to conceptualize mood instability in studies employing ecological momentary assessment (EMA). The topic addressing a methodological gap in psychiatric research. The review is generally well-structured, with a clear rationale and comprehensive search strategy, but there are areas where clarity, methodological transparency, and practical recommendations could be strengthened. The following comments are for working further on review.

  • While the introduction clearly positions mood instability as an important psychiatric construct, the research question could be framed more explicitly in terms of its intended contribution beyond descriptive synthesis.

Suggestion: Add a sentence in the final paragraph of the Introduction specifying the gap this review addresses and the specific intended output (e.g., "to develop a taxonomy for statistical methods in MI research").

  • The paper uses “mood instability” as an umbrella term, but the operational definitions of related constructs (variability, inertia, etc.) are scattered across sections. This may confuse who are not familiar with Sperry’s framework.

Suggestion: Include a dedicated table early in the Methods or Introduction with concise definitions and corresponding statistical measures from prior literature.

  • Although the search terms are described, the supplementary materials contain the full strategies rather than providing them in-text.

Suggestion: Include at least one complete search string for a database in the main Methods section to enhance reproducibility, and briefly explain any limitations in terminology coverage.

  • The authors state that risk of bias assessment was not applicable, but systematic review standards (e.g., PRISMA) generally recommend at least commenting on methodological quality.

Suggestion: Provide a short justification referencing PRISMA guidance, and possibly note any indirect bias risks (e.g., publication bias, sampling bias in EMA studies).

  • The chi-square tests are reported but not accompanied by effect sizes or a discussion of their magnitude and relevance.

Suggestion: Include Cramer’s V values or other effect size measures and comment on whether the statistically significant differences are also practically significant.

  • The Discussion calls for standardization but remains general.

Suggestion: Provide specific candidate metrics (e.g., MSSD for instability, SD for variability) and recommend scenarios in which each would be optimal, possibly in a concise decision-tree figure.

  • Figure captions could be more informative, e.g., Figure 3 could briefly interpret why MSSD dominates in clinical samples.

  • Several small typos (e.g., “uncertanties” should be “uncertainties”) should be corrected.

  • Estonia is listed twice in the study characteristics section — needs correction

  • Ensure uniform capitalization and abbreviation use for “mean squared successive differences (MSSD)” and “standard deviation (SD)”.

  • Some DOIs appear malformed with duplicated “https://doi.org/https:/” — these should be corrected for consistency.

Reviewer 3 Report

Comments and Suggestions for Authors

This manuscript titled Statistical, Conceptualisation of Mood Instability: A Systematic Review, addresses an important gap in psychiatry: the lack of consensus on how to define and statistically calculate mood instability (MI). The systematic review is timely, covers a large body of literature, and offers valuable insights into the variety of methods used. However, there are some concerns regarding clarity, methodological rigor, depth of critical synthesis, and implications for future practice.

Timely and Relevant Topic – Mood instability is increasingly recognized as a critical transdiagnostic construct across psychiatric disorders. Addressing statistical conceptualisation is highly relevant. Systematic Approach – The review followed a registered PROSPERO protocol, searched multiple databases, and included a broad range of populations (clinical and non-clinical). Inclusion of 230 studies across 24 countries provides comprehensive coverage. Clear Results Presentation – Use of PRISMA diagram, bar charts, and word cloud improves clarity.

However, some points should be considered:

  • The manuscript often conflates related constructs (variability, instability, reactivity, inertia). While this is discussed, the distinctions are not sufficiently clarified or illustrated with strong examples beyond the weather analogy. The review could better address whether different constructs require entirely separate metrics or whether they are facets of a broader latent construct.

  • Line 95–99: Exclusion of non-English papers may bias the findings, especially since EMA studies are increasingly global. Line 103–106: Hand searching seems limited to references of review papers; grey literature and dissertations were not considered.
  • Line 182–185: While three methods (MSSD, SD, MLM) dominate, the discussion does not critically evaluate why these methods are chosen or their strengths/weaknesses. Line 266–270: The statement that “sophisticated methods allow for nuanced exploration” is too general—there should be a deeper critique of statistical appropriateness. For example, MSSD captures temporal instability but assumes equal spacing of measurements, which is not always the case in EMA.

  • Line 285–289: The claim that standardisation could benefit patients and clinicians is valid but speculative. No concrete framework or recommended minimum core set of metrics is proposed. This reduces the practical impact.

  • Risk of Bias / Study Quality: Line 140–144: No risk of bias assessment was conducted. Although the outcome was "choice of statistical method," it would still strengthen the paper to evaluate study quality (e.g., EMA design quality, frequency of sampling).

  • Terminology Inconsistency: Line 180–181: 27 terminologies used to describe 15 methods reflects heterogeneity, but the review does not propose a taxonomy or glossary to harmonise usage.
  • Figures: Figures (e.g., word cloud, bar chart) are descriptive but not deeply analytical. A forest plot-style comparison of outcomes linked to methods would make the review more impactful.

  • Line 20–44 (Introduction): The review highlights conceptual confusion but does not clearly define how the authors operationally distinguish “variability” vs. “instability.” Suggest adding a table of definitions with examples.

  • Line 87–97 (Eligibility): Excluding non-English studies is a limitation that should be acknowledged explicitly in the Discussion.
  • Line 140–144 (Risk of bias): Lack of any quality assessment weakens credibility. Even a simplified framework (e.g., EMA frequency, compliance rates) could provide context.

  • Line 182–185 (Results): It would be useful to add effect sizes or associations (e.g., whether certain methods yield systematically higher MI values in clinical vs non-clinical populations).

  • Line 253–276 (Discussion): The authors note “broad agreement” on which methods match which dynamics but do not present a clear synthesis table mapping dynamics → statistical methods.

  • Line 285–289 (Clinical implications): The analogy with weight loss is helpful but oversimplified. The authors should propose a practical recommendation for clinicians (e.g., standardising on 2–3 preferred metrics).

  • Line 294–298 (Limitations): The limitation section should also acknowledge publication bias (studies with null findings may not have been published) and the evolving nature of EMA technology.

Recommendations

  1. Provide a taxonomy or glossary distinguishing variability, instability, reactivity, and inertia.
  2. Add a comparative critique of the main methods (SD, MSSD, MLM)—highlight assumptions, strengths, and limitations.
  3. Consider a risk of bias/quality framework for included studies, even if simplified.
  4. Propose a minimum core set of statistical methods to standardise future MI research.
  5. Expand clinical relevance by suggesting how clinicians or digital health apps might implement these measures.

Decision: Major Revision – The paper is highly relevant and well-structured, but it requires a deeper critical analysis, clearer definitions, and stronger clinical/practical recommendations before being suitable for publication.

Comments on the Quality of English Language

can be improved 

Reviewer 4 Report

Comments and Suggestions for Authors

To the authors,

Mood instability is an important and rather ubiquitous symptom in psychiatric disorders. Despite its obvious clinical importance, mood instability has unfortunately not received the research focus it deserves. This paper has the merit of bringing to attention the issue of inconsistencies in the statistical analysis of mood instability. It identifies an array of statistical methods used in various studies, and ultimately characterizes the trends in this analysis over the past years. However, the paper could benefit from addressing some minor issues:

  1. The first paragraph from the Introduction section should specify that mood instability is not only an attribute of affective disorders, but it is also rather frequent in psychotic disorders, as well as other personality disorders, besides borderline personality disorder.
  2. The authors might consider a clearer differentiation between the terms used and the articles analyzed. For instance, “mood variability” might refer to individual, physiological, daily variations of positive and/or negative mood, and does not necessarily imply a pathological state, whereas “mood instability” is a term usually used to describe a pathological state, where mood fluctuations and their behavioral consequences are difficult to regulate. It could be beneficial to state whether the articles referring to the physiological mood variability were excluded from the analysis.
  3. I recommend checking the numbers listed in Figure 2. PRISMA flow diagram from record identification to study inclusion: initial records identified = 926; duplicate records removed = 462, so the remaining Records screened should be 464 (not 462).
  4. Since the study’s results should be able to be verified, I suggest some form of listing some details on the 230 studies included in the review (such as a table), or adding that all data is available on request.
  5. The authors might consider expanding the Discussion section and adding some opinions and advantages/ disadvantages of the statistical methods identified, rather than just presenting the results.
  6. The article should include a Conclusion section.

Round 2

Reviewer 2 Report

Comments and Suggestions for Authors

The authors have adequately addressed all comments and provided thoughtful, well-justified responses that significantly improve the manuscript. Key revisions—including clarifying the research aim, adding a definitions table, enhancing methodological transparency, and incorporating effect sizes—have strengthened the scientific clarity and rigor of the work. Minor issues such as figure captions, typographical errors, and formatting inconsistencies were also carefully resolved. While the discussion on standardization remains cautious, the authors have reasonably explained their position and outlined future directions. Overall, the revised manuscript is scientifically sound and much improved.

Reviewer 3 Report

Comments and Suggestions for Authors

The authors have carefully and satisfactorily addressed all the comments and suggestions raised during the review process. The revisions have improved the clarity and quality of the manuscript, and no major concerns remain. In my opinion, the paper is now suitable and durable for publication in Brain Sciences Journal.